# Trends in Pregnancy-Associated Cervical Cancer in Japan between 2012 and 2017: A Multicenter Survey

**DOI:** 10.3390/cancers14133072

**Published:** 2022-06-23

**Authors:** Sayako Enomoto, Kosuke Yoshihara, Eiji Kondo, Akiko Iwata, Mamoru Tanaka, Tsutomu Tabata, Yoshiki Kudo, Eiji Kondoh, Masaki Mandai, Takashi Sugiyama, Aikou Okamoto, Tsuyoshi Saito, Takayuki Enomoto, Tomoaki Ikeda

**Affiliations:** 1Department of Obstetrics and Gynecology, Mie University School of Medicine, Tsu 514-8507, Japan; s-enomoto@med.mie-u.ac.jp (S.E.); t-ikeda@clin.medic.mie-u.ac.jp (T.I.); 2Department of Obstetrics and Gynecology, Niigata University Graduate School of Medical and Dental Sciences, Niigata 951-8510, Japan; yoshikou@med.niigata-u.ac.jp; 3Department of Obstetrics and Gynecology, Yokohama City University School of Medicine, Yokohama 236-0004, Japan; aiwata@yokohama-cu.ac.jp; 4Department of Obstetrics and Gynecology, Keio University School of Medicine, Tokyo 160-8582, Japan; mtanaka@keio.jp; 5Department of Obstetrics and Gynecology, Tokyo Women’s Medical University, Tokyo 162-8666, Japan; tabata.tsutomu@twmu.ac.jp; 6Department of Obstetrics and Gynecology, Hiroshima University Graduate School of Medicine, Hiroshima 739-0046, Japan; yoshkudo@hiroshima-u.ac.jp; 7Department of Obstetrics and Gynecology, Kumamoto University School of Medicine, Kumamoto 860-8556, Japan; kondo@kuhp.kyoto-u.ac.jp; 8Department of Obstetrics and Gynecology, Kyoto University School of Medicine, Kyoto 606-8507, Japan; mandai@kuhp.kyoto-u.ac.jp; 9Department of Obstetrics and Gynecology, Ehime University School of Medicine, Ehime 791-0295, Japan; sugiyama@m.ehime-u.ac.jp; 10Department of Obstetrics and Gynecology, The Jikei University School of Medicine, Tokyo 105-8471, Japan; aikou@jikei.ac.jp; 11Department of Obstetrics and Gynecology, Sapporo Medical University, Sapporo 060-8543, Japan; tsaito@sapmed.ac.jp

**Keywords:** uterine cervical neoplasms, pregnancy, conization, neoadjuvant therapy, trachelectomy

## Abstract

**Simple Summary:**

The available evidence does not sufficiently indicate the status of pregnancy-associated cervical cancer in Japan and the associated treatment options. This study aimed to assess the occurrence of pregnancy-associated cervical cancer, available treatments, and the impact on the mother and the unborn child. The results show that pregnancy-associated cervical cancer is occurring more frequently in Japan, and most of the patients with early-stage disease have successful deliveries, although some have preterm births. Our findings also show that pregnant women with stage IB1 cervical cancer have the option of continuing pregnancy in addition to chemotherapy, as well as removal of the cervix and surrounding tissues. Thus, the treatment approach for pregnancy-associated cervical cancer must consider the stage of the cancer and the benefits that will accrue for both the mother and the unborn child.

**Abstract:**

Large-scale data on maternal and neonatal outcomes of pregnancy-associated cervical cancer in Japan are scarce, and treatment strategies have not been established. This multicenter retrospective observational study investigated clinical features and trends in pregnancy-associated cervical cancer treatments at 523 hospitals in Japan. We included cervical cancer cases that were histologically diagnosed (between 1 January 2012, and 31 December 2017), and their clinical information was retrospectively collected. Of 40 patients diagnosed with pregnancy-associated cervical cancer at ≥22 gestational weeks, 34 (85.0%) were carefully followed until delivery without intervention. Of 163 diagnosed at <22 gestational weeks, 111 continued and 52 terminated their pregnancy. Ninety patients with stage IB1 disease had various treatment options, including termination of pregnancy. The 59 stage IB1 patients who continued their pregnancy were categorized by the primary treatment into strict follow-up, conization, trachelectomy, and neoadjuvant chemotherapy groups, with no significant differences in progression-free or overall survival. The birth weight percentile at delivery was smaller in the neoadjuvant chemotherapy group than in the strict follow-up group (*p* = 0.029). Full-term delivery rate was relatively higher in the trachelectomy group (35%) than in the other groups. Treatment decisions for pregnancy-associated cervical cancer are needed after estimating the stage, considering both maternal and fetal benefits.

## 1. Introduction

Malignant disease in pregnancy occurs in approximately 1.0 per 1000 deliveries worldwide [1]. In particular, the frequency of invasive cervical cancer in pregnancy is reported to be 1–12 per 10,000 pregnancies [1,2,3]. Pregnancy-associated cervical cancer is a problem in Japan because of the increasing proportion of births among women aged ≥ 35 years [4,5] and the increasing incidence of cervical cancer among younger women [6]. In surveys of pregnant women with malignant disease conducted in 2008 and 2014, we identified cervical cancer as the most common type of cancer during pregnancy [7,8]. However, we could not investigate the trends in pregnancy-associated cervical cancer in Japan because both surveys were single-year studies and different facilities had participated. Moreover, our previous studies focused on all types of malignancies during pregnancy and could not discuss clinical management for pregnancy-associated cervical cancer.

Previous studies have identified several aspects of pregnancy-associated cervical cancer; that is, pregnancy does not adversely affect the prognosis of cervical cancer [9,10], and chemotherapy during pregnancy results in small-for-gestational-age (SGA) newborns [11,12,13]. Additionally, in 2009, a French Working Group proposed recommendations for the management of invasive cervical cancer in pregnant women. The International Network on Cancer, Infertility and Pregnancy (INCIP) also published guidelines for the management of gynecological cancers, including cervical cancer, in 2009, and has recently updated the guidelines [14,15]. Cervical cancer treatment guidelines in various countries also now include a chapter on pregnancy-associated cervical cancer [3,16,17]. However, these guidelines for managing pregnancy-associated cervical cancer are not necessarily based on sufficient evidence. Indeed, there are no large-scale data on the long-term prognosis of pregnancy-associated cervical cancer. Therefore, this study aimed to clarify the incidence trends, treatment modalities, and maternal and neonatal outcomes in pregnancy-associated cervical cancer in Japan. 

## 2. Materials and Methods

### 2.1. Survey Protocol

This multicenter retrospective observational study was conducted at 523 secondary or tertiary care hospitals in Japan. In the primary survey, patients with pregnancy-associated cervical cancer diagnosed between 1 January 2012, and 31 December 2017, were examined. In the secondary survey, clinicopathological information was retrospectively collected from the medical records of the patients with pregnancy-associated cervical cancer. The exclusion criteria were as follows: missing data, pregnancy or postpartum not within the study period, twin pregnancy, and carcinoma in situ.

### 2.2. Data Collection and Variable Definitions

Clinical information collected in the secondary survey included age, history of pregnancy and delivery, diagnosis, primary stage, histology, gestational age at diagnosis, treatment modality, delivery method, additional treatment after delivery, and date of the last follow-up. Clinical staging was determined according to the 2008 International Federation of Gynecology and Obstetrics (FIGO) system. The percentile of birth weight was calculated based on the birth size standards by gestational age for Japanese neonates [18]. SGA was defined as a birth weight < the 10th percentile.

Pregnancy-associated cervical cancer was defined as cervical cancer histologically diagnosed during pregnancy, at delivery, or within 1 year postpartum. Histological findings included micro-invasive carcinoma or worse, except for cervical intraepithelial neoplasia. The personal information of the participants was anonymized.

Japanese law prohibits abortions after 22 weeks of gestation. Therefore, we analyzed the clinicopathological features of cervical cancer during pregnancy separately in terms of diagnosis at <22 or ≥22 weeks of gestation. Pregnancy examinations are performed at the following intervals: approximately 3 times from the first to the 11th week of pregnancy, every 4 weeks from 12 to 23 weeks, every 2 weeks from 24 to 35 weeks, and every 1 week from 36 to 40 weeks. To identify the trends in treatment choice for cervical cancer during pregnancy by stage in Japan, we categorized the primary treatment modalities administered during pregnancy as follows: strict follow-up (delayed treatment after delivery), conization, trachelectomy, and neoadjuvant chemotherapy. When both conization and trachelectomy were performed, trachelectomy was considered the main treatment; for trachelectomy and adjuvant chemotherapy, trachelectomy was considered the main treatment; and for conization and neoadjuvant chemotherapy, neoadjuvant chemotherapy was considered the main treatment. The choices of equipment for conization, trachelectomy, and chemotherapy regimens depended on the criteria at each institution. The treatment criteria for conization and trachelectomy are defined in the Japan Society of Gynecological Oncology Guidelines CQ31 and CQ32 [19,20,21,22]. The trachelectomy group underwent pelvic lymphadenectomy at initial surgery, and almost all patients underwent cerclage concurrently with surgery and postoperatively. In this case, patients were admitted to the hospital for management until delivery and were administered medication to prevent premature delivery with or without uterine contractions. In this study, there were 5 cases of vaginal trachelectomy and 13 cases of abdominal trachelectomy.

We focused on patients with stage IB1 disease who were diagnosed at a gestational age of <22 weeks because the treatment strategy for stage IB1 cervical cancer during pregnancy is controversial in the guidelines. We divided stage IB1 patients into four groups based on the abovementioned primary treatment modalities during pregnancy, and we analyzed the impact of each treatment modality during pregnancy on both maternal and neonatal outcomes.

Overall survival was defined as the time from the date of diagnosis to the date of death or the last follow-up, and progression-free survival was defined as the time from the diagnosis date to disease progression.

### 2.3. Statistical Analysis

The Cochran–Armitage trend test was used for the trend test. Continuous variables were compared among the four groups by one-way ANOVA and Tukey’s post hoc test. The Kaplan–Meier method and log-rank test were used to analyze progression-free and overall survival. All statistical analyses were performed using IBM SPSS Statistics for Windows, version 24.0 (IBM Corporation, Armonk, NY, USA) or the R program (http://www.r-project.org, access on 15 January 2019). *p* < 0.05 was considered significant.

## 3. Results

In the primary survey, 369 of 523 centers (72%) responded, of which 185 indicated that they recorded relevant cases of pregnancy-associated cervical cancer. In the secondary survey, 118 of 185 centers (64%) enrolled 307 patients (Figure 1). Of these 307 patients, 290 (203 patients diagnosed during pregnancy and 87 diagnosed postpartum) were included in the analysis according to the eligibility criteria. Of the 203 cases diagnosed during pregnancy, 163 (80%) were diagnosed at a gestational age of <22 weeks, and 40 (20%) were diagnosed at a gestational age of ≥22 weeks.

### 3.1. Trends

A total of 369,011 deliveries were recorded in the study from 2012 to 2017. When we defined the frequency of pregnancy-associated cervical cancer as the number of live birth cases in each year divided by the total number of deliveries at all the registered facilities, the frequency of cervical cancer during pregnancy (from 0.035% in 2012 to 0.050% in 2017, *p =* 0.23) did not differ significantly between 2012 and 2017, but there was an upward trend, as shown in Appendix A.

### 3.2. Patient Characteristics

Overall, 89% (260/290) of the patients had stage I disease (Table 1); among them, stage IB1 was the most common stage in both pregnancy (110/203 [54%]) and postpartum cases (37/87 [43%]). Among the patients diagnosed at ≥22 weeks of gestation, 15/40 (38%; 95%CI, 23–54%) had disease of the bulky type (i.e., tumor diameter ≥ 4 cm; stage IB2, IIA2), and the rate was significantly higher than that in the group diagnosed at <22 weeks of gestation (vs. 17/203 (8.3%; 95%CI, 5.0–13%), *p* < 0.001).

The median gestational age at diagnosis was 14 (interquartile range [IQR]; range 11–16) weeks for the group diagnosed at <22 weeks of gestation and 29.5 (IQR; range 27–34) weeks for the group diagnosed at ≥22 weeks of gestation. In the group diagnosed at ≥22 weeks of gestation, 29/40 (72%) patients were diagnosed in the third trimester. The median time of diagnosis during the postpartum period was 2 (IQR; range 2–11) months after delivery.

There were no significant differences in maternal age at diagnosis, parity, or histology between the groups diagnosed at <22 and ≥22 weeks of gestation.

### 3.3. Pregnancy Outcomes and Treatment Modality of the Women Diagnosed with Cervical Cancer at ≥22 Weeks of Gestation

Because artificial abortion is not an option in patients diagnosed at ≥22 weeks in Japan, all 40 patients continued their pregnancy until delivery. Among the patients diagnosed at ≥22 weeks of gestation, 34/40 (85%) underwent strict follow-up, 2/40 (5%) underwent neoadjuvant chemotherapy, and 4/40 (10%) underwent surgical treatment via conization (*n* = 3) or trachelectomy (*n* = 1) (Appendix A). Overall, 6/40 (15%) patients underwent treatment during pregnancy, and the median gestational age at diagnosis was 26.5 (IQR; range, 24–28) weeks. Of the 34/40 (85%; 70–94%) patients who did not undergo treatment during pregnancy, the median gestational age at diagnosis was 30 (IQR; range, 28–34) weeks. The median interval from diagnosis to delivery among the women who did not undergo treatment during pregnancy was 2 (IQR; range, 1–5) weeks (Appendix A), and 16/34 (47%) underwent concurrent radical hysterectomy and cesarean section.

### 3.4. Pregnancy Outcomes and Treatment Modalities in the Women Diagnosed with Cervical Cancer at <22 Weeks of Gestation

The pregnancy outcomes by stage in the 163 patients diagnosed at <22 weeks of gestation are shown in Appendix A. The proportion of patients who opted for artificial abortions increased with stage progression, especially from stage IB2 and above. In 47 patients who chose an artificial abortion, the median gestational ages at diagnosis and abortion were 12 (IQR; range, 9–15) and 16 (IQR; range, 12–19) weeks, respectively. Miscarriages occurred in 3.1% (95%CI, 1.0–7.0%) patients, 2 of which occurred after conization during pregnancy (surgeries at 15 and 19 weeks of gestation), 1 after trachelectomy (surgery at 17 weeks of gestation), and 2 among women who did not undergo treatment during pregnancy (Figure 1). The rate of live births tended to decline as the disease progressed.

Next, treatments administered during pregnancy by stage among the patients who chose to continue their pregnancy at <22 weeks of gestation are shown in Table 2. Most women with cervical cancer (86.5%) were treated during their pregnancy. The main treatment modality was surgery, and 80/111 (72.1%) patients underwent conization or trachelectomy.

In all, 35/37 (94.6%) patients with stage IA1 disease underwent conization, and there were no cases of maternal death during the observation period. The median gestational ages at diagnosis and delivery were 14 (IQR; 12–16) and 37 (IQR; 37–39) weeks, respectively. In patients with stage IA1 disease treated with conization, 26/35 (74.3%) continued to full term, and 9/35 (25.7%) had preterm delivery (Appendix A). Among the 9 patients with preterm delivery, 3 underwent iatrogenic preterm delivery for the purpose of early treatment of cervical cancer. In contrast, 6 patients with stage IB2 or higher disease (60.0%) were treated with neoadjuvant chemotherapy, and 2 patients (20.0%) were followed up until ≥22 weeks of gestation. On the other hand, a wide variety of treatment modalities were selected by those with stage IB1 disease.

### 3.5. Treatment for Stage IB1 Disease Diagnosed at <22 Weeks of Gestation

In 59 patients with stage IB1 cervical cancer, 26 (44%) underwent conization, 14 (23%) underwent trachelectomy, and 10 (16.9%) received neoadjuvant chemotherapy during pregnancy (Table 3). Nine patients were not treated with surgery or chemotherapy and were followed up until delivery. Table 3 shows the clinicopathological characteristics of each treatment modality during pregnancy. One patient who received adjuvant chemotherapy after trachelectomy during pregnancy was excluded from the subsequent analysis.

The median gestational age at diagnosis tended to be earlier in the trachelectomy group than in the other groups, with a median of 12.5 weeks (IQR; range 7–14), and there was a significant difference between the two groups when compared with conization (*p* = 0.020) (Figure 2A). Although trachelectomy was performed at a median of 16.0 weeks (IQR; range 15–17), neoadjuvant chemotherapy started at a median of 19.5 weeks (IQR; range, 18–21) in consideration of the fetal organogenesis period. Although there was no significant difference in the gestational weeks at delivery among the four groups (Figure 2B), the percentage of full-term births tended to be higher in the trachelectomy group (35.7%) than in the strict follow-up (11.1%), conization (15.4%), and neoadjuvant chemotherapy (0%) groups. The cause of preterm birth was iatrogenic for the purpose of early treatment of cervical cancer in all cases. The percentage of patients who underwent surgery for the treatment of cervical cancer at the same time as cesarean section was 55.6% (5/9), 76.9% (20/26), 71.4% (10/14), and 100% (10/10) in the follow-up, conization, trachelectomy, and neoadjuvant chemotherapy groups, respectively (Appendix A).

For the infants, the birth weight percentile was 72.3 (IQR; range, 67–81) and 36.5 (IQR; range, 21–47) in the follow-up and neoadjuvant chemotherapy groups, respectively, with a significantly lower percentile in the neoadjuvant chemotherapy group (*p* = 0.029) (Figure 2C). SGA was detected in 2/26 (7.8%) patients in the conization group and 1/10 (10.0%) patients in the neoadjuvant chemotherapy group.

Next, the duration of pregnancy for each treatment group was compared among the four groups (Figure 2D,E). The duration for the trachelectomy group was significantly longer than those for the follow-up (*p* = 0.019) and conization groups (*p* = 0.0031).

Finally, the oncologic outcome of each treatment modality was assessed by Kaplan–Meier survival analysis (Figure 2F,G). No significant differences in progression-free or overall survival were observed among the four groups.

## 4. Discussion

Our multicenter retrospective observational study demonstrated the frequency of cervical cancer during pregnancy between 2012 and 2017. Although no significant difference was observed, an increasing trend of cervical cancer during pregnancy was confirmed, and we speculate that low cervical cancer vaccination and cervical cancer screening uptake rates could be the underlying reasons. In Japan, repeated reports of pain and movement disorders allegedly caused by HPV vaccination led to the suspension of active vaccination in 2013, after which the vaccination rate dropped to 1% [23]. The trends of cervical cancer incidence and mortality in 31 countries in 2021, reported by Shujuan Lin et al., showed that in most countries, the incidence and mortality rates have decreased or remained unchanged over the past decade due to aggressive vaccination against cervical cancer, while in Japan, both incidence and mortality rates have been increasing [24]. Furthermore, the cervical cancer screening rate is lower in Japan than in other countries [25]. In Japan, cervical cytology is a standard examination in early pregnancy and offers the first opportunity for cervical cancer screening for many women. Since approximately 90% of pregnant women submit their pregnancy notification to the government by the 11th week of pregnancy, most pregnant women undergo screening with cervical cytology in the early stages of pregnancy. The impact of the active suspension of cervical cancer vaccine will likely continue for some time, but since active vaccination has resumed in 2022, the vaccination rate is expected to increase in the future. In addition, active cervical cancer screening should be encouraged.

In the Japanese guideline^4^ and the guideline of the International Consensus Meeting [15] on the management of cervical cancer during pregnancy, conization is recommended as the treatment for stage IA disease without lympho-vascular space invasion. In this study, 90% of the women with stage IA disease who continued pregnancy underwent conization, and no deaths occurred at the end of the study. These results support the recommendations of the guidelines. In contrast, 17.6% of the patients with stage IA1 disease treated with conization during pregnancy had preterm births, besides those that experienced iatrogenic preterm births. A meta-analysis of retrospective studies of obstetric outcomes after conization in nonpregnant patients with CIN, including stage IA1 disease, reported a preterm birth rate of 11.2% [26]. Although the number of stage IA1 patients was limited in this meta-analysis, the risk of preterm birth in patients with stage IA disease treated with conization during pregnancy may be slightly higher than that in pregnant women treated with conization prior to becoming pregnant. Therefore, careful obstetric management is required after conization during pregnancy.

The greatest concern among physicians is the appropriate treatment modality for stage IB1 disease, which is the most common stage during pregnancy [10,27]. The French guideline [9] and the guideline based on the third International Consensus Meeting [15] have established a treatment plan for FIGO 2008 stage IB1 disease with a tumor size less than 2 cm. Both guidelines recommend pelvic laparoscopic lymphadenectomy for stage IB1 disease diagnosed before 22 weeks of gestation. If lymph node metastasis is positive, interruption of the pregnancy is recommended. If lymph node metastasis is negative, delayed treatment after delivery is recommended. The guideline based on the third International Consensus Meeting [15] also recommends simple trachelectomy as another treatment option. The important point is that lymphadenectomy is not recommended after 22 weeks of gestation because a sufficient number of nodes cannot be retrieved after this gestational age [15].

Although the French guideline [9] was published in 2009, pelvic lymphadenectomy was not performed for stage IB1 disease diagnosed before 22 weeks of gestation in Japan between 2012 and 2017, excluding trachelectomy cases. The Japanese guideline [3] states that therapeutic strategies should be discussed individually for patients with stage IB1 disease diagnosed before 22 weeks of gestation, and there are four treatment modalities (follow-up, neoadjuvant chemotherapy, conization, and trachelectomy) for stage IB1 disease. No significant differences in oncologic outcomes were observed between the four treatment modalities. We believe that the prognosis did not differ due to the small number of cases and the selection bias for the cases (many cases in the trachelectomy group would have originally had their pregnancies terminated and radical hysterectomy selected, making it difficult to select follow-up). However, there were some differences in the gestational period and birth weight percentile.

Specifically, the trachelectomy group tended to show an earlier diagnosis than the other groups. Abdominal trachelectomy is often performed at approximately 16 weeks to avoid miscarriage at an early gestational age and difficulties in surgical procedures at late gestational weeks [22]. In other words, trachelectomy is a treatment modality for stage IB1 disease diagnosed at an early gestational age. The trachelectomy group also had a tendency to demonstrate a longer duration of pregnancy than the other groups. Indeed, the rate of full-term birth was higher in the trachelectomy group. This tendency might be influenced by the high radicality of trachelectomy compared to other treatment modalities. However, one miscarriage that occurred after trachelectomy was considered an iatrogenic abortion. Furthermore, the surgical techniques for trachelectomy during pregnancy are difficult to standardize and should thus be performed at high-volume centers. In addition, the neoadjuvant chemotherapy group showed lower birth weights, consistent with the findings of previous studies [11,12,13]. However, no significant difference in the frequency of SGA among the four groups was observed, probably due to the small sample size of the neoadjuvant chemotherapy group. The frequency of SGA was significantly higher in the neoadjuvant chemotherapy group compared to the other groups (OR: 5.24, 95% CI: 1.24–22.1, *p =* 0.024) when all cases were analyzed without limiting by stage (data not shown). A 20-year international cohort study of 1170 patients indicated that babies exposed to antenatal chemotherapy might be more likely to develop SGA and be admitted to the neonatal intensive care unit (NICU) than babies not exposed [11]. Therefore, the involvement of hospitals with obstetric high-care units in the management of patients with invasive cervical cancer is recommended. In addition, the platinum agent frequently used for neoadjuvant chemotherapy is classified as group 2A by the International Agency for Research Cancer (IARC) and is transferred to the fetus after maternal administration via the placenta. Therefore, the long-term prognosis for the fetus (including carcinogenesis) should be carefully considered. Whichever treatment option is chosen, it is essential that patients and their families be fully informed and involved in the treatment discussions.

This study has some limitations because of its retrospective nature, utilization of data from case series (which comprised 60%), and utilization of data from selected hospitals in Japan. Due to the retrospective study design, we were unable to update the stage of each case from the FIGO 2009 to the FIGO 2018 [28]. Nevertheless, this is the first study to investigate pregnancy-associated cervical cancer and to reveal the trends in cervical cancer during pregnancy, including treatment strategies, in Japan. These results can be useful for developing management and treatment guidelines for cervical cancer during pregnancy. Moreover, the importance of early detection of cervical cancer through appropriate cervical cancer screening and the prevention of disease onset through active vaccination against cervical cancer was reiterated.

## 5. Conclusions

Treatment decisions, including pregnancy termination for pregnancy-associated cervical cancer, should be made after estimating the stage, with careful consideration of both maternal and fetal benefits.

## Figures and Tables

**Figure 1 cancers-14-03072-f001:**
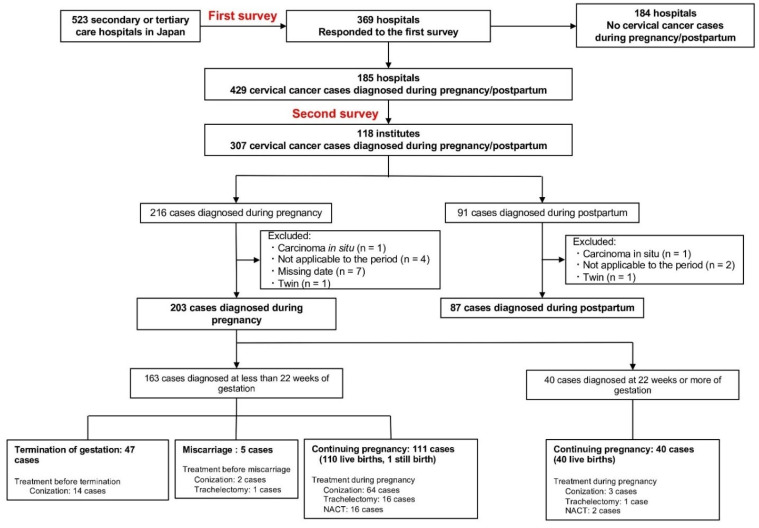
Patient inclusion flow chart. NACT, neoadjuvant chemotherapy.

**Figure 2 cancers-14-03072-f002:**
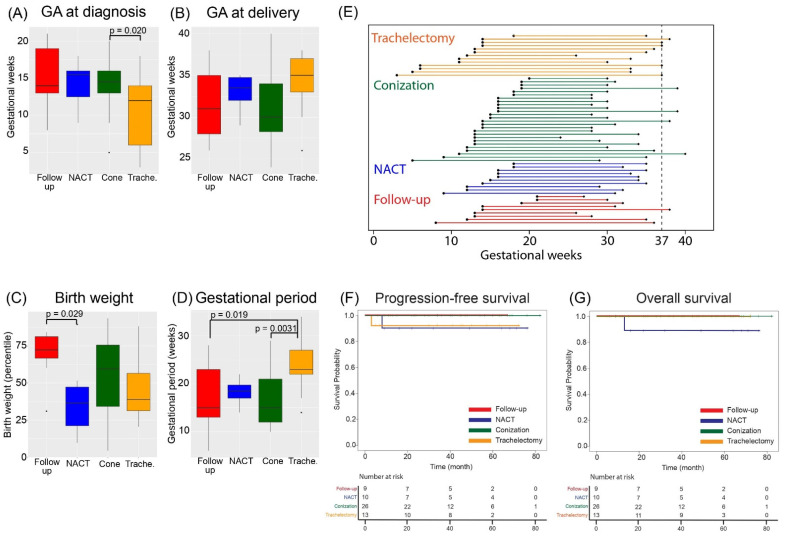
Differences in obstetrics and gynecologic outcomes among the four groups. (**A**,**B**) Comparison of gestational age at (**A**) diagnosis and (**B**) delivery. (**C**) Comparison of birth weight (percentile) at delivery among the four groups. (**D**) Comparison of the duration of pregnancy from diagnosis to delivery for each treatment group. (**E**) The duration of pregnancy from diagnosis to delivery for all cases. (**F**) The progression-free survival and (**G**) overall survival for each treatment group. GA, gestational age; NACT, neoadjuvant chemotherapy.

**Table 1 cancers-14-03072-t001:** Patient characteristics.

	Pregnancy Cases (*n* = 203)		Postpartum Cases (*n* = 87)
	<22 Weeks (*n* = 163)	≥22 Weeks (*n* = 40)	
Age at diagnosis (years)	33.0 (31–36)	32.0 (28–36)	33.0 (29–37)
Nulliparous	75 (46.0%)	15 (37.5%)	27 (31.0%)
Stage			
IA1	40 (24.5%)	3 (7.5%)	19 (21.8%)
IA2	7 (4.3%)	1 (2.5%)	4 (4.6%)
IB1	90 (55.2%)	20 (50.0%)	37 (42.5%)
IB2-IVB	26 (16.0%)	16 (40.0%)	27 (31.1%)
Histology			
Squamous cell carcinoma	114 (69.9%)	20 (50.0%)	68 (78.2%)
Adenocarcinoma	39 (23.9%)	13 (32.5%)	13 (14.9%)
Others *	10 (6.2%)	7 (17.5%)	6 (6.9%)
Gestational week at diagnosis	14.0 (11–16)	29.5 (27–34)	-
Number of months at diagnosis after delivery	-	-	2 (2–6)

Data are presented as the median (IQR; range) or *n* (%). * Others include adenosquamous carcinoma, small-cell carcinoma, large-cell carcinoma, and neuroendocrine carcinoma.

**Table 2 cancers-14-03072-t002:** Treatments administered during pregnancy by stage among patients diagnosed at a gestational age of <22 weeks who chose to continue their pregnancy (*n* = 111).

Treatment during Pregnancy	IA1 (*n* = 37)	IA2 (*n* = 5)	IB1 (*n* = 59)	IB2-IVB (*n* = 10)
Strict follow-up (*n* = 15)	1 (2.7%)	3 (60%)	9 (15.3%)	2 (20.0%) *
Conization (*n* = 64)	35 (94.6%)	2 (40%)	26 (44.1%)	1 (10.0%)
Trachelectomy (*n* = 16)	1 (2.7%)	-	14 (23.7%)	1 (10.0%)
NACT (*n* = 16)	-	-	10 (16.9%)	6 (60.0%)

Data are presented as *n* (%). NACT; neoadjuvant chemotherapy. * One person died of the disease (stage IVB squamous cell carcinoma, diagnosed at 15 gestational weeks, delivery at 26 gestational weeks, and died at 7 months postpartum).

**Table 3 cancers-14-03072-t003:** Comparison of the clinicopathological data among the four groups of patients with stage IB1 cervical cancer diagnosed at <22 weeks of gestation.

	Strict Follow-Up (*n* = 9)	Conization (*n* = 26)	Trachelectomy (*n* = 14)	NACT (*n* = 10)
Age at diagnosis (years)	37.0 (31–39)	34.0 (31–37)	32.5 (29–34)	36.0 (34–36)
Pathological tissue				
Squamous cell carcinoma	6 (66.7%)	17 (65.4%)	10 (71.4%)	7 (70.0%)
Adenocarcinoma	2 (22.2%)	8 (30.8%)	3 (21.4%)	3 (30.0%)
Others	1 (11.1%)	1 (3.8%)	1 (7.2%)	0 (0%)
Gestational age at diagnosis (weeks)	14.0 (13–19)	14.5 (13–16)	12.5 (7–14)	15.5 (13–16)
Gestational age at first treatment (weeks)	-	16.0 (15–19)	16.0 (15–17)	19.5 (18–21)
Gestational age at delivery (weeks),	31.0 (28–35)	30.0 (28–34)	35.0 (33–37)	33.5 (32–35)
Preterm delivery	8 (88.9%)	22 (84.6%)	9 (64.3%)	10 (100%)
Full term delivery	1 (11.1%)	4 (15.4%)	5 (35.7%)	0 (0%)
Mode of delivery				
Vaginal delivery	1 (11.1%)	2 (7.8%)	0 (0%)	0 (0%)
Cesarean section	8 (88.9%)	24 (92.2%)	14 (100%)	10 (100%)
Neonatal outcome				
Birth weight percentile	72.3 (67–81)	60.0 (34–76)	37.8 (31–53)	36.5 (21–47)
Small for gestational age	0 (0%)	2 (7.8%)	0 (0%)	1 (10%)
NICU admission	7 (77.8%)	21 (80.8%)	9 (64.2%)	10 (100%)

Data are presented as the median (IQR; range) or *n* (%); NACT: neoadjuvant chemotherapy, NICU: neonatal intensive care unit.

## Data Availability

The data that support the findings of this study are available from the corresponding author, E.K. (Eiji Kondo), T.E., upon reasonable request.

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
