# Peer review of "Trends in Pregnancy-Associated Cervical Cancer in Japan between 2012 and 2017: A Multicenter Survey"

_cancers, 2022, doi:10.3390/cancers14133072_

Round 1

Reviewer 1 Report

The authors conducted a nationwide cohort study that aimed to assess the occurrence of pregnancy-associated cervical cancer, available treatments, and the impact on the mother and the unborn child. Although the study has been generally well performed, there are some issues that need to be addressed.

1.       In the introduction (lines 53-56), the authors state that they expect that the incidence of pregnancy-associated cervical cancer will rise due to the increasing incidence of cervical cancer among young women. This is remarkable, as one might also argue that the improved surveillance and vaccination programs will result in a lower incidence. In their discussion (lines 250-253), the authors briefly mention that the number of Japanese women who participate in these programs is relatively low. For the reader’s understanding, it would be useful to provide percentages on how many women actively participate in these programs in Japan.

2.       Patients were excluded in case of missing data (lines 84-85). In case only a few data were incomplete, one may consider multiple imputation of missing values. This approach is known to be much more reliable than simply discarding all cases being not fully complete, as the latter may lead to selection bias. Please explain the definition of missing data in the current study and clarify why multiple imputation methods were not applied.

3.        Please explain why twin pregnancies were excluded in this study (lines 84-85). Given the rarity of both a cervical cancer diagnosis during pregnancy and a twin pregnancy, it would have been more than interesting to describe these cases separately.

4.       The frequency of cervical cancer during pregnancy was defined as the number of cervical cancer cases during pregnancy in each year divided by the total number of deliveries at all the registered facilities (lines 138-140). As pregnancies resulting in miscarriages were not taken into account in this definition, the frequency of pregnancy-associated cervical cancer might be overreported. Please address this in the text.

5.       As mentioned in the discussion, no pelvic lymphadenectomy was performed during pregnancy in patients diagnosed with FIGO stage IB1 disease at less than 22 weeks of gestational age (lines 282-289). Please provide information on lymph node status at the time of radical hysterectomy and pelvic lymphadenectomy following delivery (i.e. caesarean section). How many patients had positive lymph nodes and did require additional treatment?

6.       Does trachelectomy comprise both vaginal and abdominal trachelectomy? Was a cervical cerclage applied in all patients to prevent preterm birth? Please comment.

7.       One patient who received adjuvant chemotherapy after trachelectomy was excluded from subsequent analysis (lines 211-212). Please provide details on the reason for administering adjuvant chemotherapy in this patient. Moreover, please clarify whether chemotherapy was administered during pregnancy or postpartum, as this is not clear.

8.       The percentages of preterm birth were 88.9% and 84.6% in patients with FIGO stage IB1 disease who underwent follow-up and conization, respectively (Table 3). Yet, the range of gestational age at delivery were 28-35 and 28-34 weeks for both groups, respectively, which are all below 37 weeks and thus premature. Shouldn’t these percentages therefore be changed into 100% preterm birth? Please explain.

9.       Strikingly, patients who underwent trachelectomy had a relatively lower risk of preterm birth compared to patients who underwent conization. How can these differences be clarified?

Author Response

Comments and Suggestions for Authors

Dear Reviewer 1

The authors conducted a nationwide cohort study that aimed to assess the occurrence of pregnancy-associated cervical cancer, available treatments, and the impact on the mother and the unborn child. Although the study has been generally well performed, there are some issues that need to be addressed.

Dear Reviewer:

Thank you very much for your comments concerning our manuscript. The comments were valuable and very helpful in revising and improving our manuscript. We have studied the comments carefully and made corrections, which we hope will receive your approval. The revised sections are highlighted in yellow in the revised manuscript.

Comment 1.                                                                         In the introduction (lines 53-56), the authors state that they expect that the incidence of pregnancy-associated cervical cancer will rise due to the increasing incidence of cervical cancer among young women. This is remarkable, as one might also argue that the improved surveillance and vaccination programs will result in a lower incidence. In their discussion (lines 250-253), the authors briefly mention that the number of Japanese women who participate in these programs is relatively low. For the reader’s understanding, it would be useful to provide percentages on how many women actively participate in these programs in Japan.

➡Response;

Thank you for your question.

In Japan, the HPV vaccine was approved for clinical use in 2009 and has been available at public expense for girls aged 13–16 years since 2010, followed by routine vaccination for those aged 12–16 years since April 2013. However, after the spring of 2013, repeated reports of adverse events, such as pain and movement disorders allegedly caused by HPV vaccination, led to the discontinuation of active recommendation of HPV vaccination for girls in June of the same year. As a result, new immunization coverage in 2013 declined from a routine annual immunization coverage of approximately 70% to 1% for 12-year-old girls and 3–9% for 13-year-old girls [reference]. Thus, the coverage rate is 60–70% for those aged 23–28 (as of 2022), who were eligible for immunization at the time. However, in the other age groups, the vaccination rate is lower, at less than 20%.

 According to the Ministry of Health, Labour, and Welfare (MHLW), the adverse events of 2013 had no association with the HPV vaccine, and the decision was made to resume the vaccination recommendation in April 2022. A relief program was established to allow women born between 1997 and 2005, who missed the opportunity for free vaccination during the vaccination suspension period, to receive free vaccination for three years starting April 2022. Interest in the cervical cancer vaccine is growing, and the vaccination rate is currently increasing.

We have revised parts of the Discussion (Page 9, Lines 282-289) in response to your important comments. Please check it.

References

Morimoto, A.; Ueda, Y.; Egawa-Takata, T.; Yagi, A.; Terai, Y.; Ohmichi, M.; Ichimura, T.; Sumi, T.; Murata, H.; Kanzaki, H.; et al. Effect on HPV Vaccination in Japan Resulting from News Report of Adverse Events and Suspension of Governmental Recommendation for HPV Vaccination. Int. J. Clin. Oncol. 2015, 20, 549–555, doi:10.1007/s10147-014-0723-1.

Tanaka, Y.; Ueda, Y.; Egawa-Takata, T.; Yagi, A.; Yoshino, K.; Kimura, T. Outcomes for Girls Without HPV Vaccination in Japan. Lancet Oncol. 2016, 17, 868–869, doi: 10.1016/S1470-2045(16)00147-9

Ueda, Y.; Enomoto, T.; Sekine, M.; Egawa-Takata, T.; Morimoto, A.; Kimura, T. Japan’s Failure to Vaccinate Girls Against Human Papillomavirus. Am. J. Obstet. Gynecol. 2015, 212, 405–406, doi:10.1016/j.ajog.2014.11.037.

Comment 2.

Patients were excluded in case of missing data (lines 84-85). In case only a few data were incomplete, one may consider multiple imputation of missing values. This approach is known to be much more reliable than simply discarding all cases being not fully complete, as the latter may lead to selection bias. Please explain the definition of missing data in the current study and clarify why multiple imputation methods were not applied.

➡Response;

Thank you for your remarks. In this study, cases in which the number of weeks of delivery was unknown and cases in which the prognostic outcome was unknown because the patient moved out of the hospital and could not be followed were excluded as missing dates. We will consider multiple imputation methos in the next study.

Comment 3.                                                                     Please explain why twin pregnancies were excluded in this study (lines 84-85). Given the rarity of both a cervical cancer diagnosis during pregnancy and a twin pregnancy, it would have been more than interesting to describe these cases separately.

➡Response;

Thank you for the comment. Twin pregnancies were excluded because of the high risk of preterm delivery and low birth weight, as well as the effects of treatment, such as surgery and chemotherapy, and the difficulty in determining the effects.

Comment 4.                                                                       The frequency of cervical cancer during pregnancy was defined as the number of cervical cancer cases during pregnancy in each year divided by the total number of deliveries at all the registered facilities (lines 138-140). As pregnancies resulting in miscarriages were not taken into account in this definition, the frequency of pregnancy-associated cervical cancer might be overreported. Please address this in the text.

➡Response;

We appreciate the reviewer’s positive and constructive suggestions on our manuscript.

We have corrected eFigure 1 after receiving similar remarks from other reviewers. The number of registered cases of cervical cancer during pregnancy divided by the total number of deliveries at all registered facilities in each year was shown as the incidence rate of cervical cancer during pregnancy in Japan, but since the registered cases included miscarriage and abortion cases, we excluded those cases and reanalyzed only the live birthcases. Although an increasing trend was observed, the difference was no longer significant, and the text was changed. Thank you very much for your kind attention.

  We have revised the following sentence in the Introduction section Page 2, Lines 55-56; Pregnancy-associated cervical cancer is a problem in Japan, and the following sentence in the Results section Page 4, Lines 152-157; When we defined the frequency of pregnancy-associated cervical cancer as the number of live birth cases in each year divided by the total number of deliveries at all the registered facilities, the frequency of cervical cancer during pregnancy (from 0.035% in 2012 to 0.050% in 2017, P=.23) did not differ significantly between 2012 and 2017, but there was an upward trend, as shown in Figure S1.

Comment 5.                                                                        As mentioned in the discussion, no pelvic lymphadenectomy was performed during pregnancy in patients diagnosed with FIGO stage IB1 disease at less than 22 weeks of gestational age (lines 282-289). Please provide information on lymph node status at the time of radical hysterectomy and pelvic lymphadenectomy following delivery (i.e. caesarean section). How many patients had positive lymph nodes and did require additional treatment?

➡Response;

Thank you for your comment. We apologize for the confusing text. Patients who underwent tracheostomy during pregnancy had lymph node dissection at the same time: for trachelectomy for cervical cancer 1B1 diagnosed at <22 weeks, information about the lymph nodes was available in 11 of 14 cases, and none of these cases showed lymph node metastasis. For the other treatment groups, lymph node dissection in some cases was performed at or after delivery, but the presence or absence of lymph node metastasis is unknown because the related data were not included in this study. We apologize for this point.

We have added the following sentence in the Materials and Methods section Page 3, Line 115-116: The trachelectomy group underwent pelvic lymphadenectomy at initial surgery.

Comment 6.                                                                        Does trachelectomy comprise both vaginal and abdominal trachelectomy? Was a cervical cerclage applied in all patients to prevent preterm birth? Please comment.

➡Response;

Thank you for your question.                                                         We have added the following sentence in the Materials and Methods section Page 3, Lines 119-120: In this study, there were 5 cases of vaginal trachelectomy and 13 cases of abdominal trachelectomy.

Comment 7.                                                                       One patient who received adjuvant chemotherapy after trachelectomy was excluded from subsequent analysis (lines 211-212). Please provide details on the reason for administering adjuvant chemotherapy in this patient. Moreover, please clarify whether chemotherapy was administered during pregnancy or postpartum, as this is not clear.

➡Response;

 Thank you for the comment. In this case, trachelectomy was performed at 19 weeks, and then adjuvant chemotherapy was performed before delivery. We excluded this case as the treatment effect of surgical treatment or chemotherapy is duplicated, and it is difficult to judge. We have revised the following sentence in the Results section Page 7, Lines 234-236: One patient who received adjuvant chemotherapy after trachelectomy during pregnancy was excluded from the subsequent analysis.

Comment 8.                                                                       The percentages of preterm birth were 88.9% and 84.6% in patients with FIGO stage IB1 disease who underwent follow-up and conization, respectively (Table 3). Yet, the range of gestational age at delivery were 28-35 and 28-34 weeks for both groups, respectively, which are all below 37 weeks and thus premature. Shouldn’t these percentages therefore be changed into 100% preterm birth? Please explain.

➡Response;                                                                                  Thank you for your comment. The range of gestational age at delivery is described at the bottom of the Table and is the IQR value. The minimum-maximum range was 26-38 weeks for the follow up group and 24-40 weeks for the conization group.

Comment 9.                                                                  Strikingly, patients who underwent trachelectomy had a relatively lower risk of preterm birth compared to patients who underwent conization. How can these differences be clarified?

➡Response;

Thank you for the comment. In this study, we cannot give a definite answer regarding the presence or absence of cerclage for conization because we did not receive full data. In the trachelectomy cases, additional research showed that almost all patients underwent cerclage at the same time as surgery and postoperatively. In this case, the patients were admitted to the hospital for management until delivery and were given medication to prevent premature delivery for 7–14 days with or without uterine contractions. It is conceivable that this could have led to favorable results.

We have added the following sentence in the Materials and Methods section Page 3, Lines, 116-119: … almost all patients underwent cerclage concurrently with surgery and postoperatively. In this case, patients were admitted to the hospital for management until delivery and were administered medication to prevent premature delivery with or without uterine contractions.

Reviewer 2 Report

x

Author Response

Comments and Suggestions for Authors

Dear Reviewer 2,

Thank you for reviewing our manuscript.

Reviewer 3 Report

Cervical cancer in Pregnancy is rare and precious study topic. I suggest major revision

1.     “Incidence of pregnancy associated cervical cancer is expected to increase in Japan’ I want additional explanation about bias in Japan about this population. What’s current guideline and insurance coverage for Pap and/or HPV typing, Colposcopy biopsy, Tumor marker, Imaging workup like MRI or CT during pregnancy and postpartum? Are they delayed diagnosis just in pregnancy time? Is there standard referral system from maternal fetal medicine to gynecologic oncologist during pregnancy? And is there risk factor associated finding about pregnancy and cervical cancer besides smoking or persistent HPV infection and delayed colposcopy biopsy? What’s age distribution of cervical cancer especially 20,30 and 40 in Japan regardless of pregnancy? Is HPV vaccination rate low among all reproductive age? How about other malignancy incidence like breast cancer during pregnancy?

2.     Additional information about “Conization & Trachelectomy” is needed. What’s criteria of conization & trachelectomy?. Is there pathologic result like size, depth, margin, lymphovascular space invasion report? Is there mandatory preventive procedure like cerclage, preterm medication after conization/trachelectomy? Was Cold knife incision or Coagulation allowed during procedure? Any special precaution about anesthesia and postop medication?

3.     Additional information about “Strict follow up group” How often Pap and/or HPV, Colposcopy, Tumor marker, Imaging modality?

4.     What is the guessing cause of birth weight problem like SGA in this study group?

5.     Is there additional explanation about “No significant differences in progression-free or overall survival were observed among the four groups” Does author conclude “Strict follow up has same result with surgical intervention and chemotherapy during pregnancy in cervical cancer Ib1”?

Author Response

Comments and Suggestions for Authors

Dear Reviewer 3:

Thank you very much for your comments concerning our manuscript. The comments were valuable and very helpful in revising and improving our manuscript. We have studied the comments carefully and made corrections, which we hope will receive your approval. The revised sections are highlighted in yellow in the revised manuscript.

Comment1.                                                                                                           

“Incidence of pregnancy associated cervical cancer is expected to increase in Japan’ I want additional explanation about bias in Japan about this population.

➡Response:

We appreciate the reviewer’s positive and constructive suggestions on our manuscript.

We have corrected eFigure 1 after receiving similar remarks from other reviewers. The number of registered cases of cervical cancer during pregnancy divided by the total number of deliveries at all registered facilities in each year was shown as the incidence rate of cervical cancer during pregnancy in Japan, but since the registered cases included miscarriage and abortion cases, we excluded those cases and reanalyzed only the live birth cases. Although an increasing trend was observed, the difference was no longer significant, and the text was changed. Thank you very much for your kind attention.

  We have revised the following sentence in the Introduction section Page 2, Line 55-56: Pregnancy-associated cervical cancer is a problem in Japan, and the following sentence in the Results section Page 4, Line 152-157 When we defined the frequency of pregnancy-associated cervical cancer as the number of live birth cases in each year divided by the total number of deliveries at all the registered facilities, the frequency of cervical cancer during pregnancy (from 0.035% in 2012 to 0.050% in 2017, P=.23) did not differ significantly between 2012 and 2017, but there was an upward trend, as shown in Figure S1.

(Comment1.)

What’s current guideline and insurance coverage for Pap and/or HPV typing, Colposcopy biopsy, Tumor marker, Imaging workup like MRI or CT during pregnancy and postpartum? Are they delayed diagnosis just in pregnancy time? Is there standard referral system from maternal fetal medicine to gynecologic oncologist during pregnancy?

➡Response;

Thank you for your insightful comment. In Japan, approximately 90% of pregnancies are notified within 11 weeks (about 5% within 12–19 weeks), and pregnant women who have notified the government of their pregnancy receive a free pregnancy examination ticket, which includes Papanicolaou pap smear as an early pregnancy examination. Therefore, most pregnant women receive Papanicolaou pap smear at the 10-week mark of pregnancy. If the Papanicolaou pap smear reveals a problem, obstetric guidelines indicate that, in principle, the same measures should be taken as for non-pregnant women, and colposcopy and biopsy are also performed during pregnancy with insurance coverage. In addition, pregnancy-associated cervical cancer cases are referred from primary to higher tertiary facilities as soon as they are diagnosed. Therefore, table 1 shows cervical cancer cases diagnosed in pregnancies reported at less than 22 weeks (n=163); the median number of weeks was 14 weeks. However, 40 cases were diagnosed after 22 weeks, and it is possible that these were sampling errors or that the women were not seen by a health care provider early in their pregnancy.                                  

We have added the following sentence in the Discussion section Page 9, Lines 289-294:

In Japan, cervical cytology is a standard examination in early pregnancy and offers the first opportunity for cervical cancer screening for many women. Since approximately 90% of pregnant women submit their pregnancy notification to the government by the 11th week of pregnancy, most pregnant women undergo screening with cervical cytology in the early stages of pregnancy.

(Comment 1.) 

And is there risk factor associated finding about pregnancy and cervical cancer besides smoking or persistent HPV infection and delayed colposcopy biopsy?

➡Response;

  Thank you for your comment. This study did not investigate smoking history, HPV infection rates, or vaccination, and therefore cannot be determined. We apologize.

(Comment 1.)

What’s age distribution of cervical cancer especially 20,30 and 40 in Japan regardless of pregnancy?

➡Response;

Thank you for your comment. According to the 2018 data, the age group-specific incidence rates per 100,000 population for cervical cancer were 0.7 in the 20–24 years age group, 5.2 in the 25–29 years age group, 18.1 in the 30–34 years age group, 26.4 in the 35–39 years age group, 27.8 in the 40–45 years age group, and 29.2 in the 45–49 years age group.

(Comment 1.)

Is HPV vaccination rate low among all reproductive age?

➡Response;

 Thank you for your comment. In Japan, the HPV vaccine was approved for clinical use in 2009 and has been available at public expense for girls aged 13–16 years since 2010, followed by routine vaccination for those aged 12–16 since April 2013. However, after the spring of 2013, repeated reports of adverse events, such as pain and movement disorders allegedly caused by HPV vaccination, led to the discontinuation of active recommendation of HPV vaccination for girls in June of the same year. As a result, new immunization coverage in 2013 declined from a routine annual immunization coverage of approximately 70% to 1% for 12-year-old girls and 3–9% for 13-year-old girls [reference]. Thus, the coverage rate is 60–70% for those aged 23–28 years (as of 2022), who were eligible for immunization at the time. However, in the other age groups, the vaccination rate is lower, at less than 20%.

 According to the Ministry of Health, Labour, and Welfare (MHLW), the adverse events of 2013 had no association with the HPV vaccine, and the decision was made to resume the vaccination recommendation in April 2022. A relief program was established to allow women born between 1997 and 2005, who missed the opportunity for free vaccination during the vaccination suspension period, to receive free vaccination for three years starting April 2022. Interest in the cervical cancer vaccine is growing, and the vaccination rate is currently increasing.

We have revised parts of the Discussion (Page 9, Line 282-288) in response to your important comments. Please check it.

Reference

Morimoto, A.; Ueda, Y.; Egawa-Takata, T.; Yagi, A.; Terai, Y.; Ohmichi, M.; Ichimura, T.; Sumi, T.; Murata, H.; Kanzaki, H.; et al. Effect on HPV Vaccination in Japan Resulting from News Report of Adverse Events and Suspension of Governmental Recommendation for HPV Vaccination. Int. J. Clin. Oncol. 2015, 20, 549–555, doi:10.1007/s10147-014-0723-1.

Tanaka, Y.; Ueda, Y.; Egawa-Takata, T.; Yagi, A.; Yoshino, K.; Kimura, T. Outcomes for Girls Without HPV Vaccination in Japan. Lancet Oncol. 2016, 17, 868–869, doi: 10.1016/S1470-2045(16)00147-9

Ueda, Y.; Enomoto, T.; Sekine, M.; Egawa-Takata, T.; Morimoto, A.; Kimura, T. Japan’s Failure to Vaccinate Girls Against Human Papillomavirus. Am. J. Obstet. Gynecol. 2015, 212, 405–406, doi:10.1016/j.ajog.2014.11.037.

Hanley, S.J.; Yoshioka, E.; Ito, Y.; Kishi, R. HPV Vaccination Crisis in Japan. Lancet. 2015, 385, 2571, doi:10.1016/S0140-6736(15)61152-7

(Comment1.)                                                                         How about other malignancy incidence like breast cancer during pregnancy?

➡Response;

Thank you for your question. Surveys of pregnant women with malignant disease conducted in 2014 in Japan (although the surveys were single-year studies and different facilities had participated in the research) found the following: Of the 510 medical centers, 411 (81%) responded to the survey. There were 215,372 deliveries and 189 incidents (0.09%) of malignant disease in pregnancy. Of the 189 patients with malignancy, 157 detailed responses about the patients were received. The most frequently encountered cancer types were cervical cancer (36%), breast cancer (24%), and ovarian cancer (15%). During the 2 years after delivery, 15 patients (1 with breast cancer, 2 with ovarian cancer, 3 with hematologic malignancy, 4 with intestinal cancer, and 5 with others) died of the disease; most of them had advanced disease.

Reference

Kobayashi, Y.; Tabata, T.; Omori, M.; Kondo, E.; Hirata, T.; Yoshida, K.; Sekine, M.; Itakura, A.; Enomoto, T.; Ikeda, T. A Japanese Survey of Malignant Disease in Pregnancy. Int. J. Clin. Oncol. 2019, 24, 328–333, doi:10.1007/s10147-018-1352-x.

Comment 2.

 Additional information about “Conization & Trachelectomy” is needed. What’s criteria of conization & trachelectomy?

➡Response;

Thank you for your important comment. Japanese cervical cancer guidelines recommend conization during pregnancy for definitive diagnosis if stage IA is suspected during pregnancy. The recommended time for conization is at around 14–15 weeks of pregnancy, and coin-biopsy should be performed considering the risk of bleeding and preterm delivery.                                                               The following is a single institution's standard for trachelectomy.                       

1 Strong desire to continue the pregnancy and raise the child                               2 Ability to perform the procedure at 15-18 weeks gestation                                3 1A2 and 1B1 stage                                                                  4 Tumor diameter less than 2 cm                                                       5 No obvious lymph node metastasis                                                    6 Histology must be SCC, Adc, or adenosquamous carcinoma                              7 Tumor edge must be 1.5 cm away from the endocervical opening.                          However, because this is a retrospective observational study, none of the above criteria were standardized in this study, and the final decision is at the physician's discretion.

We have added the following sentence in Materials and Methods section Page 3, Lines 114,115;

The treatment criteria for conization and trachelectomy are defined in the Japan Society of Gynecological Oncology Guidelines CQ31 and CQ32.

(Comment 2.)

Is there pathologic result like size, depth, margin, lymphovascular space invasion report? Is there mandatory preventive procedure like cerclage, preterm medication after conization/trachelectomy?

➡Response;

Thank you for your comment. There are some missing cases, but we have the results of additional research on size, margin, and lymphovascular space invasion, which we will show you in the below table. (data not shown in Manuscript)

In the case of conization, cerclage may be performed at the same time as surgery to prevent postoperative preterm labor, according to Japanese cervical cancer guidelines, but there is still no certain opinion regarding its effectiveness, and its implementation should be at the physician's discretion. The present study does not have a complete set of data regarding the presence or absence of cerclage for conization; thus, it is not possible to give a definite answer.

Regarding the trachelectomy cases, additional research showed that almost all patients underwent cerclage at the same time as surgery. Patients were admitted to the hospital for management until delivery, and were given medication to prevent premature delivery for 7–14 days with or without uterine contractions.

We have added the following sentences in the Materials and Methods section Page 3, Line 116-119: … almost all patients underwent cerclage concurrently with surgery and postoperatively. In this case, patients were admitted to the hospital for management until delivery and were administered medication to prevent premature delivery with or without uterine contractions.

(Comment 2.) 

Was Cold knife incision or Coagulation allowed during procedure? Any special precaution about anesthesia and postop medication

➡Response;

Thank you for the comment. Although there may be cases of cold knife incision, we did not investigate the details of the surgical technique, including anesthesia, and cannot make a judgment. We apologize for the inconvenience.

 Comment 3.                                                                       Additional information about “Strict follow up group” How often Pap and/or HPV, Colposcopy, Tumor marker, Imaging modality?

➡Response;

Thank you for the comment. We did not investigate the tests and management procedures performed after the cervical cancer diagnosis in the Strict follow up group. We apologize for the absence of the data. In general, the Japanese guidelines stipulate that the Pregnancy checkup intervals should be approximately 3 times between the first and 11th week of pregnancy, every 4 weeks from 12 to 23 weeks, every 2 weeks from 24 to 35 weeks, and every 1 week from 36 to 40 weeks. For the cases included in the present study, it is likely that the examination was conducted at the same frequency or more frequently. The details of the examination are basically the physician's choice.   

We have added the following sentences in the Materials and Methods section Page 3, Lines 102-105: Pregnancy examinations are performed at the following intervals: approximately 3 times from the first to the 11th week of pregnancy, every 4 weeks from 12 to 23 weeks, every 2 weeks from 24 to 35 weeks, and every 1 week from 36 to 40 weeks.

Comment 4.  

What is the guessing cause of birth weight problem like SGA in this study group?

➡Response;

Thank you for your question. As noted in the Discussion (Page 10, lines 350-352) , the frequency of SGA was significantly higher in the neoadjuvant chemotherapy group compared to the other groups (OR: 5.24, 95% CI: 1.24–22.1, P= .024) when all cases were analyzed without limiting by stage. In utero chemotherapy has been reported to cause SGA in many cases, and we are considering the effects of neoadjuvant chemotherapy in this study.

Comment 5.                                                                         Is there additional explanation about “No significant differences in progression-free or overall survival were observed among the four groups” Does author conclude “Strict follow up has same result with surgical intervention and chemotherapy during pregnancy in cervical cancer Ib1”?

➡Response;

Thank you for your important point.                                        We believe that the prognosis did not differ due to the small number of cases and the selection bias for the cases (many cases in the Trachelectomy group would have originally had their pregnancies terminated and radical hysterectomy selected, making it difficult to select follow-up).                                                                Although the results showed no significant difference in prognosis, trachelectomy has the advantage of longer duration of pregnancy. In addition, although the follow-up group has no problem of weight corrected for the number of weeks, a large percentage of the patients are born prematurely; thus, we believe that the problem of premature births will arise.

We have added the following sentence in the Discussion section Page 10, Lines 331-334: We believe that the prognosis did not differ due to the small number of cases and the selection bias for the cases (many cases in the trachelectomy group would have originally had their pregnancies terminated and radical hysterectomy selected, making it difficult to select follow-up).

Reviewer 4 Report

This is very large retrospective review of cervical cancer in pregnancy in Japan. It provides evidence of an increasing incidence of this problem, and looks at treatment options for various stages of disease and gestational ages. I have the following concerns:

(i) Line 95. You state: Histological findings included microinvasive carcinoma or worse, except for cervical intraepithelial neoplasia. The CIN is superfluous and should be omitted

(ii) Figure 1. Why did you eliminate the two cases of twins?

(iii) Line 163. Of the patients >22 weeks, the median age at diagnosis was 26.5 weeks, but only 5% received NACT. Many centres would use more NACT to allow the pregnancy to continue. You should comment on the Japanese philosophy in relation to NACT

(iv) Line 251. You mention that HPV vaccination rates in Japan decreased due to reported complications. You should expand on the reported complications and whether or not vaccination rates have increased again, because you advocate for active vaccination in line 324

(v) Line 327. You should emphasis that the advantages and disadvantages of the various treatment options should be discussed with both parents

Author Response

Dear Reviewer 4,

Comments and Suggestions for Authors

This is very large retrospective review of cervical cancer in pregnancy in Japan. It provides evidence of an increasing incidence of this problem, and looks at treatment options for various stages of disease and gestational ages. I have the following concerns:

Dear Reviewers:

Thank you very much for your comments concerning our manuscript. The comments were valuable and very helpful in revising and improving our manuscript. We have studied the comments carefully and made corrections, which we hope will receive your approval. The revised sections are highlighted in yellow in the revised manuscript.

Comment 1.                                                                       Line 95. You state: Histological findings included microinvasive carcinoma or worse, except for cervical intraepithelial neoplasia. The CIN is superfluous and should be omitted

➡Response;

We have removed “CIN” from the text. Thank you for pointing that out.

Comment 2.                                                                     Figure 1. Why did you eliminate the two cases of twins?

➡Response;

Thank you for your comment. Twin pregnancies were excluded because of the high risk of preterm delivery and low birth weight, as well as the effects of treatment, such as surgery and chemotherapy, and the difficulty in determining the effects.

Comment 3.                                                                       Line 163. Of the patients >22 weeks, the median age at diagnosis was 26.5 weeks, but only 5% received NACT. Many centres would use more NACT to allow the pregnancy to continue. You should comment on the Japanese philosophy in relation to NACT

➡Response;

 Thank you very much for your important remarks. Of the 34 patients diagnosed at 22 weeks or more, the median number of weeks of diagnosis was 30 weeks (IQR; 28–34) and the median number of weeks of delivery was 34.5 weeks (IQR; 32–37) for those who chose to follow up. Since intact survival is higher at births over 28 weeks in Japan [reference], it is possible that these patients chose early delivery after diagnosis rather than in utero treatment. The two patients who chose NACT during pregnancy were diagnosed at 22 and 28 weeks and delivered at 30 and 36 weeks, respectively; both had advanced stage IB2 or higher cancer, one was diagnosed at 22 weeks, and there was concern about prematurity of the infant, which may have led them to choose NACT during pregnancy to prolong the gestation period. Another possible reason is that NACT during pregnancy is not a common treatment method in Japan, and the number of facilities performing it are limited.  

Reference

Kusuda, S.; Fujimura, M.; Uchiyama, A.; Totsu, S.; Matsunami, K. Trends in Morbidity and Mortality Among Very-Low-Birth-Weight Infants From 2003 to 2008 in Japan. Pediatr. Res. 2012, 72, 531–538, doi: 10.1038/pr.2012.114

Maeda K. Highly Improved Perinatal States in Japan. J. Obstet. Gynaecol. Res. 2014, 40, 1968–1977, doi: 10.1111/jog.12485

Yorifuji, T.; Naruse, H.; Kashima, S.; Murakoshi, T.; Kato, T.; Inoue, S.; Kawachi, I. Trends of Preterm Birth and Low Birth Weight in Japan: A One Hospital-Based Study. BMC Pregnancy Childbirth. 2012, 12, 162, doi: 10.1186/1471-2393-12-162

Comment 4.                                                                       Line 251. You mention that HPV vaccination rates in Japan decreased due to reported complications. You should expand on the reported complications and whether or not vaccination rates have increased again, because you advocate for active vaccination in line 324

➡Response;

Thank you for your important suggestion.

In Japan, the HPV vaccine was approved for clinical use in 2009 and has been available at public expense for girls aged 13–16 years since 2010, followed by routine vaccination for those aged 12–16 years since April 2013. However, after the spring of 2013, repeated reports of adverse events, such as pain and movement disorders, allegedly caused by HPV vaccination led to the discontinuation of active recommendation of HPV vaccination for girls in June of the same year. As a result, new immunization coverage in 2013 declined from a routine annual immunization coverage of approximately 70% to 1% for 12-year-old girls and 3–9% for 13-year-old girls [reference]. Thus, the coverage rate is 60–70% for those aged 23–28 years (as of 2022), who were eligible for immunization at the time. However, in the other age groups, the vaccination rate is lower, at less than 20%.

 According to the Ministry of Health, Labour, and Welfare (MHLW), the adverse events of 2013 had no association with the HPV vaccine and the decision was made to resume the vaccination recommendation in April 2022. A relief program was established to allow women born between 1997 and 2005, who missed the opportunity for free vaccination during the vaccination suspension period, to receive free vaccination for three years starting April 2022. Interest in the cervical cancer vaccine is growing, and the vaccination rate is currently increasing.

We have revised parts of the Discussion (Page 9, Lines 282-288) in response to your important comments. Please check it.

References

Morimoto, A.; Ueda, Y.; Egawa-Takata, T.; Yagi, A.; Terai, Y.; Ohmichi, M.; Ichimura, T.; Sumi, T.; Murata, H.; Kanzaki, H.; et al. Effect on HPV Vaccination in Japan Resulting from News Report of Adverse Events and Suspension of Governmental Recommendation for HPV Vaccination. Int. J. Clin. Oncol. 2015, 20, 549–555, doi:10.1007/s10147-014-0723-1.

Tanaka, Y.; Ueda, Y.; Egawa-Takata, T.; Yagi, A.; Yoshino, K.; Kimura, T. Outcomes for Girls Without HPV Vaccination in Japan. Lancet Oncol. 2016, 17, 868–869, doi: 10.1016/S1470-2045(16)00147-9

Ueda, Y.; Enomoto, T.; Sekine, M.; Egawa-Takata, T.; Morimoto, A.; Kimura, T. Japan’s Failure to Vaccinate Girls Against Human Papillomavirus. Am. J. Obstet. Gynecol. 2015, 212, 405–406, doi:10.1016/j.ajog.2014.11.037.

Hanley, S.J.; Yoshioka, E.; Ito, Y.; Kishi, R. HPV Vaccination Crisis in Japan. Lancet. 2015, 385, 2571, doi:10.1016/S0140-6736(15)61152-7.

Comment 5.                                                                       Line 327. You should emphasis that the advantages and disadvantages of the various treatment options should be discussed with both parents

➡Response;

Thank you very much for pointing this out. We have emphasized it as such. 

We have added the following sentence in the Discussion section Page 10, Lines 361,362: Whichever treatment option is chosen, it is essential that patients and their families be fully informed and involved in the treatment discussions.

Round 2

Reviewer 1 Report

The article can be published

Reviewer 2 Report

x

Reviewer 3 Report

Well corrected,Thanks